# "There Is No Official Divorce Ritual in the Church"—Challenging a Mantra by Ritual Design

**Bernhard Lauxmann**

Department of Practical Theology and Psychology of Religion, University of Vienna, 1010 Wien, Austria; bernhard.laumann@univie.ac.at

**Abstract:** End-of-relationship rituals are not respected within Protestant churches. Therefore, many church websites emphasize that there are no official divorce rituals to this day. This is despite the fact that there are numerous impulses for divorce rituals, some of which are also firmly established in church practice. Practitioners who devote themselves to these rituals often have to be innovative. This article presents three unpublished drafts for separation rituals, which are understood as expressions of ritual design. Experiences from academic teaching show that the field divorce rituals is particularly suitable for initiating learning processes on ritual design, a skill of great importance for future pastors. Although divorce rituals are unpopular now, they are likely to become part of the standard repertoire of churches soon.

**Keywords:** pastoral training; ritual design; divorce; practical theology; liturgics; worship service; occasional church service

---

In memory of Peter Karner (14 May 1937–21 December 2022), pioneer in the field of church divorce rituals, who passed away during the publication process of this article

## 1. Introduction

This article sets out to trace the ambivalent role of divorce rituals in ecclesiastical and theological contexts and to explain why the idea that there are no official divorce rituals in the church is still fixed in the minds of many Protestants in Germany, Austria, and Switzerland. Although divorce rituals are still widely considered a no-go in Protestant churches and lead a shadowy existence, there are young pastors and aspiring Protestant theologians who thrive in their role as ritual designers and are making innovative contributions to the implementation of ritual church services in the context of divorce. In doing so, they are building on the groundwork of pioneering men and women in the field. Three previously unpublished designs are presented to a wider public in this article.

This article is structured as follows: In Section 1, I show that divorce rituals occupy a special role when it comes to the distribution of attention in theology and the church. In Section 2, I review church websites that promote the view that there are no official church divorce rituals and ask how this notion took hold. In Section 3, I discuss the preliminary work that has been conducted in the field of divorce rituals both in the media and public sphere ("pioneers") as well as hidden from view ("guerrilla practitioners"). In Section 4, I outline three concrete proposals for ritual practice in this field—a field that is both controversial and often neglected—made by a younger generation of theologians, before discussing some points that arise in connection with these proposals. The *extensive appendix* provides a detailed outline of the structure of the three theologically responsible ritual practice proposals in the context of separation/divorce that are discussed in this article. This article is written from the perspective of a Protestant theologian, who researches and teaches at the University of Vienna, Austria.

## 2. The Unequal Focus on Rituals in Theology and the Church

Protestant theologians and pastors are constantly dealing with rituals. But not all rituals are of the same interest to them—at least that is what debates in German-speaking theology and church culture suggest. Here, there are at least *three different fields of ritual practice* that differ significantly in their popularity and fascination. *Disaster rituals* are cherished and positively valued. *Regular church rituals* (occasional services and traditional Sunday worship services) are often the focus of attention simply because they have been for quite some time. Mostly, they are assessed in a balanced or ambivalent way. *Separation or divorce rituals* have a poor standing and comparatively attract little attention. Frequently, they are roundly rejected or ignored in dominant ecclesiastical and theological circles. In the following, the differences between these three fields of rituals will be made clear.

### 2.1. Disaster/Crisis Rituals: Big Love for the Big Public

*Disaster* or *crisis rituals* often receive a great deal of media attention and figure most prominently in current debates. Church leaders are proud of these rituals, gladly devote themselves to them and put them in the spotlight. Theological research, too, has recently focused on such rituals with considerable enthusiasm (Kranemann and Menzel 2021; Stetter 2022; Lauxmann 2022b).

The findings are interesting: In Germany, Austria, and Switzerland, the churches are accorded a decisive role in ritual crisis management and official commemoration rituals in connection with catastrophic events continue largely to be performed as religious services (Stetter 2022). The leading function of church personnel, the symbolic meaning of church spaces, and the ritual competence of pastors—all this is made visible in disaster and crisis rituals for the public, and it is missed in the very few cases where churches do not play a special role in today's disaster rituals (Kranemann and Menzel 2021). Such a key role is no doubt very flattering for people professionally involved in the church, especially in times of de-traditionalization, "de-churching", self-secularization, religious indifference, and other erosion phenomena (Pickel 2017; Ziebertz 2004; Huber [1998] 1999). Disaster and crisis rituals seem destined to be the focal point for a public theology or a public church (Körtner et al. 2020)—the public significance of churches and theologies becomes immediately apparent here. In other words, theologians and clergy fearing for their relevance and who in many cases have lapsed into self-pity in the face of declining membership and loss of reach (Gutmann and Peters 2021) seem to have found a field of religious–ritual practice where "their" church can accomplish something that exceeds the capacity of the religiously neutral state (Stetter 2022). The more public, the better—that, at any rate, seems to be their current motto.

### 2.2. Regular Church Services: Ambivalence towards Established Ritual Traditions

When it comes to regular church rituals, the mood is considerably less euphoric. The standard repertoire of church rituals undoubtedly includes *occasional services* (i.e., baptisms, confirmations, marriages, and funerals) and *traditional worship services* in the church year, especially on Sundays.

The fact that the Sunday worship service is a ritual was one of the fundamental insights of the early 20th century. Many practical theologians continue to describe the traditional Sunday worship service as a ritual (Graupner 2019; Haunerland 2021; Weyel 2011; Josuttis 1991; Jetter 1986). Some scholars describe these worship services as transitory rites (Weyel 2011), others prefer speaking of rituals of interaction (Walthert 2017; Walti 2016, 2017). Rooted in in the church year with its ceremonial cycle, worship services are, above all, calendrical rituals, which admittedly—in the sense of a ritual complex—implement various ritual dimensions (cf. Spiro [1982] 2020a).

Occasional church services are life-cycle-rituals (cf. Spiro [1982] 2020b), traditionally understood as rites of passage (van Gennep [1909] 2005). Scholars also call them rituals at the margins of the church, because they are often accessed by people who disrupt traditions out of a lack of familiarity with them (Wagner-Rau [2000] 2008). More recently, they have

also been characterized as "rituals in motion" (Wagner-Rau and Handke 2019). From the perspective of most of those involved, they are family rituals—and are therefore also perceived as such in the theory of practical theology (Morgenthaler 2022).

Today, both types of rituals—occasional services and regular worship services—are often continued from a sense of tradition by church professionals who often feel that they are producing prefabricated rituals on an assembly line, finding it more and more difficult themselves (Engemann 2022). In the face of apparent stasis, even pastors who want to shape the future of the church defeatedly reach such conclusions as this: "I am convinced that very many people are looking for an experience of transcendence, for peace or meaning. For beauty, eternity or love. *They don't find it in our standard program*" (Herzig et al. 2021, p. 39).

Theological research consistently deals with these traditional rituals—productively and critically. However, scholars are also aware that their immanent standard ritual program has been *floundering* for some time now (Hirsch-Hüffell 2021). Christians who shape their way of being a Christian with practices of *Doing Singularity* find it increasingly difficult to relate to these rituals. Their needs and demands for life-related rituals strongly challenge the understanding of traditional rituals within the church's standard repertoire (Bühler et al. 2020; Lauxmann 2023). Regular services are too attached to "the general" and can hardly keep up with the change of social logics in society, which today are mainly oriented towards "the unique" (Reckwitz [2017] 2019; Lauxmann 2021; Cordemann 2021). Only time will tell if the booming agencies for occasional services (e.g., *St. Moment* in Hamburg; *Segensbüro* in Berlin; *Segens_Raum* in Vienna) can really counteract this by performing special religious events (baptism festivals, river baptisms, wedding festivals, etc.) and breaking with some standard programs that have long been unquestionably established within the church in favor of increased customer orientation (Fendler 2019).

So, in the overall view, this group of rituals (occasional services and Sunday worship services) is much less popular and much more criticized within current debates than the previous group (disaster/crisis rituals). Nevertheless, they are very present, both in the daily work of pastors and in academic reflection on Christian religious practice. They have come to stay—and are stable enough to change, adapt, and move on.

*2.3. Divorce Services and Separation Rituals: Aversion to New Ritual Designs*

Separation or divorce rituals undoubtedly occupy the most difficult position. Following semantic paths laid out by official church websites (see Tables 1 and 2), they are a kind of occasional ritual that does not officially exist and should only be practiced—if at all—in consultation with a responsible parish pastor: locally, unofficially, and principally in non-public areas. For most church officials, separation or divorce rituals seem to belong to the realm of the private or semi-private. The position is clear: pastors, theologians, and church members should not make a big fuss about them. The key argument is also clear: there is no official divorce ritual in the Protestant Church. This argument is constantly emphasized. Therefore, these rituals can be understood as not respected rituals within church life. With few exceptions (esp. Bianca 2015), they are also largely ignored by theological research.

**Table 1.** Statements on divorce rituals from official local church/parish websites.

| Community/Parish | Statement/Position * |
| --- | --- |
| Protestant Schoenau Parish in Mannheim (Germany) | "… *when we get divorced*, is there a church ritual *for this?* *An important* task of pastors ('*Eine wichtige* Aufgabe von Pfarrern und Pfarrerinnen') is to accompany people in crisis situations. Please *ask/contact our* pastor ('*sprechen Sie* bitte *unseren* Pfarrer *an*') to find out which forms and rituals make sense and are possible *for you. So far*, there is no official divorce ritual in the Protestant Church, *but it is currently being discussed in various places*". (Ev-Schönau 2022) |
| Protestant Lutheran Parish Altenholz (Germany) | "*Is there a church ritual on the occasion of a divorce? An important* task of pastors [*is*] to accompany ('*Eine wichtige* Aufgabe von Pastorinnen und Pastoren [ist] die Begleitung …') people in crisis situations. Please *discuss/clarify with the* pastor ('*sprechen Sie* bitte mit *dem* Pastor *ab*') which forms and rituals make sense and are possible *for you. So far*, there is no official divorce ritual in the Protestant Church. *However, in our congregation there is an openness to let married couples go their separate ways reconciled through a ritual after a pastoral conversation*". (Ev-Altenholz 2022) |
| German Working Group for Protestant Pastoral Care of the Deaf (Germany) | "*When a marriage breaks up … An important* task of pastors ('*Eine wichtige* Aufgabe von Pastorinnen und Pastoren') is to accompany people in crisis situations. Please *ask/contact your* pastor ('*sprechen Sie* bitte *bei Ihrer* Pfarrerin oder *Ihrem* Pfarrer *an*') which forms and rituals make sense and are possible *for you. So far*, there is no official divorce ritual in the Protestant church, *although it is currently being discussed in various places*". (DAFEG 2022) |
| Protestant Parish Bellheim-Knittelsheim (Germany) | "*Is there a church ritual on the occasion of a divorce?* There is no official divorce ritual in the Protestant Church, *even though it is currently being discussed in various places*. ('…, auch wenn darüber derzeit an verschiedenen Stellen diskutiert wird')" (Ev-Bellheim 2022) |
| Protestant Lutheran Laetare Parish in Munich (Germany) | "*Is there a church ritual on the occasion of a divorce? An important* task of pastors ('*Eine wichtige* Aufgabe von Pfarrerinnen und Pfarrern') is to accompany people in crisis situations. Please *ask/contact your* pastor ('*sprechen Sie* bitte *bei Ihrer* Pfarrerin oder *Ihrem* Pfarrer *an*') which forms and rituals make sense and are possible *for you*. There is no official divorce ritual in the Protestant Church *so far, but we offer you individual possibilities. Please contact us!*" (Ev-Laetare 2022) |
| Protestant Lutheran Parish in Hasbergen (Germany) | "*Is there a church ritual on the occasion of a divorce? One* task of the pastor ('*Eine* Aufgabe des Pastors/der Pastorin …') is to accompany people, *including* in crisis situations ('*auch in Krisensituationen*'). Please *ask/contact your* pastor ('*sprechen Sie* bitte *bei ihrem* Pastor/*Ihrer* Pastorin *an*') which forms and rituals make sense and are possible. *So far*, there is no official divorce ritual in the church". (Ev-Hasbergen 2022) |
| Protestant Twelve Apostles Parish (Germany) | "*Is there a church ritual on the occasion of a divorce?* Accompanying people in crisis situations is *an important* task *of the church* ('Menschen in Krisensituationen zu begleiten ist *eine wichtige Aufgabe der Kirche*'). Which forms and rituals make sense and are possible *should be discussed/talked about* with *the* pastor ('*sollten Sie mit dem* Pfarrer oder *der* Pfarrerin *besprechen*'). There is no official divorce ritual in *the* church". (Ev-Apostel 2022) |
| (Lutheran and Reformed) Protestant Parish in Baden, Pfaffstätten and Heiligenkreuz (Austria) | "*Is there a church ritual on the occasion of a divorce? One* task of the pastor ('*Eine* Aufgabe des/r Pfarrer/in') is to accompany people, *including* in crisis situations ('*auch in Krisensituationen*'). Please *discuss/clarify with your* pastor ('*sprechen Sie* bitte mit *ihrem/ihrer* Pfarrer/in *ab*') which *concrete* forms and rituals make sense and are possible. *So far,* there is no official divorce ritual in the church". (Ev-Baden 2022) |
| (Lutheran) Protestant Parishes Gloggnitz and Naßwald (Austria) | "*Is there a church ritual on the occasion of a divorce? One* task of the pastor ('*Eine* Aufgabe des/r Pfarrer/in …') is to accompany people, *including* in crisis situations ('*auch in Krisensituationen*'). Please *discuss/clarify with your* pastor ('*sprechen Sie* bitte mit *ihrem/ihrer* Pfarrer/in *ab*') which *concrete* forms and rituals make sense and are possible. *So far*, there is no official divorce ritual in *our* church ('in *unserer* Kirche')". (Ev-Gloggnitz 2022) |

**Table 1.** *Cont.*

| Community/Parish | Statement/Position * |
|---|---|
| (Lutheran) Protestant Parish Voitsberg (Austria) | *"Is there a church ritual on the occasion of a divorce?* *One* task of *the* pastor ('*Eine* Aufgabe *des* Pfarrers'... ) is to accompany people, *including* in crisis situations ('auch in Krisensituationen'). Please *discuss/clarify with your* pastor ('sprechen Sie bitte mit *ihrem* Pfarrer *ab*') which *concrete* forms and rituals make sense and are possible. So far, there is no official divorce ritual in the church". (Ev-Voitsberg 2022) |
| Reformed Parish Thalheim (Switzerland) | *"Is there a church ritual on the occasion of a divorce?* *An important* task of pastors ('*Eine wichtige* Aufgabe von Pfarrerinnen und Pfarrern') is to accompany people in crisis situations. Please *ask/contact your* pastor ("sprechen Sie bitte bei Ihrer *Pfarrerin* oder *Ihrem* Pfarrer *an*") which forms and rituals make sense and are possible *for you. So far*, there is no official divorce ritual in *the Protestant Reformed Church ('in der Evangelisch-reformierten Kirche')*". (Ev-Thalheim and Huber 2022) |

* Words or units of meaning that illustrate textual variants have been *italicized* by the author of this article.

**Table 2.** Statements regarding divorce rituals on regional/national church websites.

| Church/Organization | Statement/Position |
|---|---|
| Protestant Church in Germany (EKD) | *"We have separated.* Is there a church ritual on the occasion of a divorce? Accompanying people in crisis situations is *one of the core tasks* of pastors ('gehört zu den Kernaufgaben von Pfarrerinnen und Pfarrern'). *It is best to clarify with your* pastor ('klären Sie am besten mit *Ihrer* Pfarrerin oder *Ihrem* Pfarrer') which forms and rituals make sense and are possible *for you. So far*, there is no official divorce ritual in the *Protestant* Church. *However, some parishes offer services with liturgical texts for separated and divorced people. Ask about this locally*". (EKD 2022) |
| Protestant Church of Anhalt (ELA) | *"Is there a church ritual on the occasion of a divorce?* *One* of the pastor's tasks ('*Eine* Aufgabe des/r Pastors/in... ' is to accompany people, *including* in crisis situations ('..., *auch* in Krisensituationen'). *Please ask/contact your* pastor ('sprechen Sie bitte *bei ihrer/m* Pastor/in *an*') which forms and rituals make sense and are possible. *So far*, there is no official divorce ritual in the *Protestant* Church, *although this is currently being discussed in various places*". (ELA 2022) |
| Bremen Protestant Church (BEK) | *"Is there a church ritual on the occasion of a divorce?* There is no official divorce ritual in the *Protestant* Church. *However,* pastors accompany people *as pastoral counselors* in crisis *and separation* situations ('Pastoren/Pastorinnen begleiten Menschen aber *als Seelsorger* und [sic!] Krisen- und Trennungssituationen'). *You can discuss/talk with him/her* ('können Sie mit Ihm/Ihr besprechen') which *rituals and forms* make sense and are possible". * (BEK 2019) |
| Protestant Church in Austria (EKiÖ) | *"Is there a church ritual on the occasion of a divorce?* *One* of the pastor's tasks ('*Eine* Aufgabe des/r Pfarrer/in... ') is to accompany people, *including* in crisis situations ('... *auch* in Krisensituationen'). *Please discuss/clarify with your* pastor ('sprechen Sie bitte mit ihrem/ihrer Pfarrer/in *ab*') which concrete forms and rituals make sense and are possible. *So far*, there is no official divorce ritual in the church". ** (EKiÖ 2022) |
| evangelisch.de (EKD article on GEP news platform) | Is there a church ritual on the occasion of a divorce? *An important* task of pastors ('*Eine wichtige* Aufgabe von Pastorinnen und Pastoren... ') is to accompany people in crisis situations. *Please ask/contact your* pastor ('sprechen Sie bitte *bei Ihrer* Pfarrerin oder *Ihrem* Pfarrer *an*') which forms and rituals make sense and are possible for you. *So far*, there is no official divorce ritual in the *Protestant* Church, *although this is currently being discussed in various places.* (EKD and GEP [2014] 2022) |
| evangelisch.de (EPD article by Barbara Driessen on GEP news platform) | *"The Protestant Church encourages pastors to write liturgical texts for the occasion and to make appropriate offerings to affected congregation members* [... ]. There is no official divorce ritual in the *Protestant* Church. *The former Protestant bishop Margot Käßmann recommends, for example, communion together as a church ritual after a separation*". *** (Driessen and EPD [2012] 2022) |

* *Note:* The statement of the Bremen Protestant Church (BEK) is *no longer online* since a website relaunch. The question regarding divorce rituals no longer appears on the website and is, therefore, no longer answered. ** The statement of the Protestant Church in Austria (EKiÖ) *has not changed* since at least 2018, although their understanding of a church wedding has changed, a new official wedding agenda was installed by the General Synod of the Lutheran Church (A.B.) in June 2021 (EKiÖ-AB 2021) and—as a result—*several formulations in the Q&A section were recently adjusted.* *** Although *evangelisch.de* is an official church website, the quote is an excerpt from an *article* written by a specific author for the *epd* (Protestant Press Service). This distinguishes it from an official church statement.

Although the Protestant regional churches in Germany act independently in many respects and often take up the templates, recommendations, or guidelines set by the church umbrella organization, EKD, and, only to a limited extent, the statements concerning the possibility of a church divorce ritual on their websites are surprisingly uniform and stereotypical. Interested individuals find little information regarding divorce rituals,

encountering, instead, a mantra—"*there is no official divorce ritual in the (protestant) church*"—that has been repeated across various church websites and repeated for almost a decade now (see Tables 1 and 2).

The mantra can be found on far more than 100 church or parish websites today. In Austria, it is used not only by the national church but also by various parishes in all provinces: e.g., *Baden* (Ev-Baden 2022), *Gloggnitz* (Ev-Gloggnitz 2022), and *Voitsberg* (Ev-Voitsberg 2022). An information brochure on Reformed marriage, published by the parish of *Thalheim*, proves that the phrase is also encountered in (Reformed) Switzerland (Ev-Thalheim and Huber 2022). In Germany, instances are legion, covering the entire territory of the EKD and its member churches from A (*Altenholz* in Schleswig-Holstein; Ev-Altenholz 2022) to Z (*Zwölf Apostel* in Berlin; Ev-Apostel 2022). Throughout the German-speaking world, one encounters the same refrain: *there is no official divorce ritual in the church.* This speaks volumes about the importance that churches attach to such a ritual today. Given the mass of evidence, it seems useful to clarify the origin of this mantra.

### 3. Mantra-like Distancing: Who Started It All?

In 2012, Barbara Driessen wrote a text for the Protestant Press Service (EPD) entitled "Even a Divorce Needs Rituals". In this article, Driessen appears to have formulated the key sentence cited above: "There is no official divorce ritual in the Protestant church" (Driessen and EPD [2012] 2022). This phrase seems to have gained wide currency since 2014, especially due to the short paragraphs on the subject of divorce rituals published in the Q&A sections on regional and national church websites (see Table 2). The variations and elaborations cannot conceal that the vast majority of texts are based on a common template. But which? Can it be dated? Does it really go back only as far as Barbara Driessen's text, and what about the context in which it was written? For some time, I have been wondering who started all this: who was the first to spread the idea that there are no (official) divorce rituals—through a simple website text? In addition, I also ask myself: since when are rituals performed by official ministers, and often even secured by resolutions of the presbyteries, not official rituals of the church?

Exegetically trained observers will note that the original wording probably goes back to an EKD text that is no longer directly available. The textual variants delivered by the Protestant Church of Anhalt (ELA 2022) and by the Protestant Church in Austria (EKiÖ 2022) are probably quite close to the original wording. Numerous parish websites, many of which are no longer up to date, also provide us with a very similar or even the same version of the text. The Protestant Church of Anhalt (with its comparatively original text version) still *explicitly* refers to the EKD as the source for its version (ELA 2022). However, the EKD's own website contains only a more detailed, obviously extended and adapted version of the text (EKD 2022), which is certainly not the original template. Rather, the EKD seems to have adapted its answer to the question of divorce rituals on the official website over time. The fact that the EKD text on the news platform *evangelisch.de* has also received an update supports this assumption: the text found there first went online in 2014, was adapted in 2018, and remains online in the updated version at the time of writing.

Barbara Driessen may have transmitted a core sentence of the original text template in 2012 by writing that "there is no official divorce ritual in the Protestant Church". Obviously, this is the key phrase as it is also encountered in the numerous later and wordier variants. But was she really drawing on an existing church (template) text? Or (conversely) did her article cause the churches to take up the issue of divorce rituals on their websites? Has, in other words, the EKD taken up *her* text and *her* formulation? This may be the case. Driessen, whom I contacted as part of the research for this article, writes: "Unfortunately, I no longer remember the details. But I suspect that the content of the sentence goes back to Pastor Armin Beuscher from Cologne, whom I quote in my article. In this sentence I presumably summarized his train of thought somewhat more briefly and concisely. In principle, your thesis is quite possible. For I have witnessed it several times that certain paraphrases or

quotations that are disseminated via news agencies actually catch on, simply on account of their high circulation".

The oldest evidence of the dominant and much reproduced text version with its typical structure and its larger text volume is the EKD article first published in 2014 on *evangelisch.de* is—unfortunately, today this EKD article is only available in an updated version. If one does not want to consider this lost text version as the original template for everything that follows, then one must conclude that the TAQ (*terminus ante quem*) for the original text template is 2014. Its TPQ (*terminus post quem*) may date back as far as 2007/2008.

In 2007/2008, the divorce of the prominent bishop Margot Käßmann became known and was discussed both in the media and within the church (Scheidung der Bischöfin Käßmann 2007). On the occasion of her divorce, people remembered that Käßmann had already demanded a church divorce ritual in a bishop's report in 2001 (Pfister and Fleischhauer 2013), in a sensational question to the synod (Bianca 2015, p. 341) and in press statements (EPD 2000); in 2008, many observers were wondering if and how she would ritually process her own divorce. Therefore, it is no coincidence that Driessen explicitly addresses Käßmann's proposal on the question of church divorce rituals in her 2012 article. Furthermore, in 2008, the Council of the EKD commissioned a document intended to correct the church's idea of the family; Käßmann explicitly calling it "a guide to how we deal with families who no longer fit the classical image" (Pfister and Fleischhauer 2013). In 2013, the highly controversial orientation guide was published (Kirchenamt der EKD 2013). After all, it seems obvious that the original text, variations on which quickly spread across church websites, belongs to this phase of church debate about socio-moral guiding principles between 2008 and 2014. If there is an original text template and the 2014 version of the EKD article on *evangelisch.de* (EKD and GEP [2014] 2022) is not the original, it would have to be dated between 2008 and 2014. In my view, it is probable that the original was written between 2011 and 2014—the connection between Driessens's EPD text (June 2012) and the EKD's Q&A (June 2014) is too compelling.

## 4. Challenging the Mantra: Pioneers and Guerilla Practitioners

Theological research on divorce rituals operates in a difficult field. Divorce rituals do not have much of a lobby. The little research that does exist has recently drawn attention to the *pioneers* who have contributed to developing divorce rituals, especially in an early phase (Lauxmann 2020; Bianca 2002, 2015)—among them the Austrian theologian and Reformed pastor *Peter Karner*, who was regional superintendent of the Protestant Church H. B. (Reformed) in Austria from 1986 to 2004. Karner deserves credit for being the first in the German-speaking world to bring the topic of church divorce rituals to the fore. His so-called *divorce sermon*—scheduled for broadcast on 27 October 1979—was shamefully cancelled at the behest of the ecclesiastical authorities (Bianca 2015, pp. 297–99; Lauxmann 2020).

From an international perspective, Karner's advance came relatively late: already in 1966, in the USA, the pastor, health educator, and clinical sexologist *Ronald Mazur* called for a divorce ritual as an expression of a healing pastoral practice and published his reflections in a journal (Mazur 1966), without, however, having presented a concrete draft (Bianca 2015, pp. 165–66). In addition, *Rudolph W. Nemser*, pastor of the Fairfax Unitarian Chuch even performed such a divorce ritual in 1966, about which he also published (Nemser 1967; cf. Bianca 2002, pp. 97–83; 2015, pp. 166–67). At approximately the same time that there was a controversial dispute regarding Karner's *divorce sermon* within the Austrian broadcasting community, the first denominationally published proposal for a divorce ritual in the United States was being intensively debated in the United Methodist Church. Although it was published only in a supplementary volume (UMC 1976), this move was probably even more of a scandal as Karner's. The Methodist Church ultimately refrained from a more official publication because of the vehemence of the debates in congregations, popular media, and faculties (cf. Bianca 2002, p. 82). This brief insertion makes it clear: not only in Austria, but elsewhere as well, initial pushes for church-based divorce rituals were thwarted. Sometime

later, having attained a higher position within the church hierarchy, the Austrian pioneer Karner—apparently, an incorrigible man—also produced a written draft for a divorce ritual (Karner 1998).

Another person who is called a pioneer has already been the subject of this article: the German theologian and Lutheran pastor *Margot Käßmann*, bishop of the Church of Hanover (ELKH) from 1999 to 2010, has tirelessly pointed out the relevance of church divorce rituals and made concrete proposals for their regular anchoring in the ritual repertoire of the church. According to Bianca's assessment, starting in the 2000s, Käßmann thereby reignited debates regarding divorce rituals (Bianca 2002; 2015, pp. 341–43). The position that there was no official divorce ritual—initially implied in the church's inaction but from 2014 onwards spelled out across a seemingly endless number of church websites—was questioned by pioneers who were convinced that such a ritual ought to be created, and sooner rather than later. There were more such pioneers than one might think, as well as those within the church working to establish a ritual practice for separated or divorced couples, or for those currently going through a process of separation and seeking help.

It is astonishing how many people come forward when an article on the subject appears in a wide-reach church medium, as was the case when I published a small reflection entitled "Unwelcome Rituals" (German *Unliebsame Rituale*) in the Protestant Church newspaper for Austria (Lauxmann 2020). These readers told me that they had already offered such rituals— some as early as the 1990s—and had done so out of pastoral commitment, and performing rituals of their own devising with one or with both (former) partners present. Although these rituals often had comparatively few participants—meaning that the degree of actual publicity was usually rather small—they were nevertheless designed as complete liturgical services. Moreover, like any service, from a theological perspective they had a claim to publicity—however small it might be. They were not "private masses" to be ignored. I like to think of them as "guerrilla rituals", "guerilla pioneers", or "guerilla practitioners".

One such guerrilla practitioner was the theologian, children's book author, and prison pastor *Christine Hubka*, who is familiar with the situation in the Austrian church. She matter-of-factly told that she had performed rituals with divorced people as a natural expression of her official pastoral mission, making my question whether that had been allowed seem almost inappropriate. Such pastoral guerrilla initiatives, on the one hand, and the high-profile public advances of pioneers, on the other, have undoubtedly had an impact. Practical theological research has discovered the topic, and the next generation of theologians can build on what has been done in the past—informed, with academical support, and without the feeling of being alone in the field.

Today, Christine Hubka is well over 70 years old, Margot Käßmann and Andrea Marco Bianca over 60. Peter Karner died during the final publication phase of this article—at the age of 85. His (Reformed) church honors him as a pugnacious churchman and as a politically thinking person who was not afraid to swim against the tide or to provoke scandals (EPD-Ö 2022). In different ways, they all have made great contributions to the field of ecclesiastical divorce rituals. Hubka, Käßmann and Bianca still continue to do so today. Now, however, the focus will be on the younger generation, which can build on their insights and initiatives. These are people who were not yet born when Karner first pushed the issue, were not yet adults when Käßmann revived the debates, and can only wonder about static text websites that stay unchanged for a decade and (implicitly) claim to make authoritative statements regarding the official or nonofficial status of rituals.

## 5. Open Liturgical Resistance: New Ritual Ideas from Young Theologians

*5.1. How Younger Generations Deal with Rituals*

The younger generation is familiar with a concept of ritual that is less concerned with social order and repeatability, one in which tradition means not so much a commitment to the old but rather a transformative process of development. Today, it is widely recognized in ritual studies that late modern rituals are definitely not old-style, invariable, and unalterable but highly mobile, dynamic, and changing (Hoondert and Post 2021; Brosius et al. 2013;

Kapferer 2006; Jungaberle and Fletcher 2006; Bell [1992] 2009). The fluidity of rituals and their constant change is directly experienced by the younger generation in their lives, even in their church lives. It follows that younger theologians and future pastors find it *easier to play the role of ritual designer* than previous generations did. This is clearly indicated by the three liturgical designs below for rituals in the context of divorce:

- Pastor Stephan Heinlein, in his mid-forties, has continued to adapt a draft ritual of his own devising because there were no usable templates. His field-tested design (see Section 5.1; Appendix B) combines classical elements of a traditional occasional worship service with his own ideas and texts—the ritual proposal is designed *for one person* who gives the former partner back into God's hand, revokes his/her own vows, and ritually separates from his/her partner.

Some theology students went even further in a liturgical seminar held at the University of Vienna in the winter semester 2019/20, an advanced course that was built around a concrete and realistic (but fictitious) scenario. Didactically, the seminar was built around the case study concept developed at Harvard Business School. Students were asked to develop a solution to the unsolved problem of divorce rituals within the Austrian Protestant Church. It was assumed that, after a lengthy debate, the liturgical committee of the Protestant Church decided to address the issue and to provide working material, not least for further debate in the synod. The students created liturgical drafts on the basis of this imaginary but plausible case (see Appendix A for details).

The proposals made by the students were impressive:

- A first group of students, most of them in their 20s, adapted the worship format to such a degree that it has taken on a rather new, unconventional shape. They rearranged elements of different origins and condensed various symbolic actions in their ritual proposal—the result is the "*Three Pillar Model*" (see Section 5.2; Appendix C), a ritual for *both former partners* and for any (potential) children from this and/or other relationships.
- In their role as ritual designers, a second group (of the same age) devised *a comprehensive and novel ritual complex* combining various workshops, individual ritual actions, personal exercises, and worship services into an overarching religious concept. The result is called "*Exploring New Paths. Protestant Retreats for Separated and Divorced People*" (see Section 5.3; Appendix D). The proposal is *for a group of people with experience of separation or divorce* who—without knowing each other beforehand—reflect together on their experiences and express these ritually. The design provides for recurring times in which individuals, supported by a group, deal with their own experience, their distinctive case.

All three drafts are discussed in more detail (see Sections 5.2–5.4), and their structure is laid out in the appendices (see Appendices B–D). However, it is already clear from the different target group definitions in the brief descriptions how multifaceted designs for a divorce ritual are today. Because there are no ecclesiastically binding guidelines for such cases, no restrictions regarding the layout/structure/goal, etc., of a draft for a divorce ritual that must be followed. This freedom is fully exercised today by a generation that has "ritual design" in its blood. The field of divorce rituals is still open and will therefore inevitably attract further ritual design experiments. Unlike in the context of disaster rituals, it is possible here to experiment, combine, and create something relatively new through hybridizing and curating practices. One important point of disaster and crisis rituals is to maintain existing lines, to continue well-rehearsed traditions, and to ensure that there are recognizable features. Because *routines provide security* in a crisis, any adaptation of such rituals must be developed very slowly and carefully. Theologians largely agree that trying to create unique, special rituals would be completely wrong in this context (Lauxmann 2022b). Since the opposite is true for divorce rituals, the field offers great opportunities for contemporary ritual design: divorce rituals cannot rely on established lines of tradition. They thus challenge ritual design—especially when they take place with a time lag from

the civil divorce and, thus, stage the conclusion of a longer process. They end an unfinished situation and—as a Gestalt therapist would say—*close an open Gestalt* (cf. Klessmann 2017, p. 93). Future pastors can learn a lot in the field of divorce rituals, especially concerning the design of contemporary rituals that are no longer oriented towards "the general". The traditional rituals (worship services and occasional services) of the church function as a general background infrastructure on which relatively new and special ritual designs are produced.

In times when (despite a decline in divorce figures due to the coronavirus pandemic) every third marriage is divorced, such designs are indispensable resources for a sustainable and healing pastoral ministry: the overall divorce rate was 36.1% in Austria (Kaindl and Schipfer 2022, p. 11) and 39.9% in Germany (Statista Research Department 2022) in 2021. That divorces have become "normal" cases of late-modern relationship culture is also evidenced by the large number of consensual divorces: the percentage of consensual divorces in Austria has remained relatively constant at between 86% and 88% in recent years (Kaindl and Schipfer 2022, p. 11). In Germany, both partners agreed to the divorce petition in 88.9% of cases in 2021 (Destatis 2022). Undoubtedly, designs for divorce rituals meet a reality in which they are needed. Because divorces not only mark transitions into new life situations and roles but are also often experienced as particularly critical periods of life ("crisis situations"), their pastoral accompaniment is rightly understood by the churches as an important task, which is indicated not least by the churches' websites (see Tables 1 and 2). Those who, in accordance with the church's mission, ritually accompany people in such crisis situations not only fulfill their ministry but also support people in overcoming profound life crises.

### 5.2. The "Liturgy of Letting Go" (Germany 2018)

Pastor Stephan Heinlein developed his *Liturgy of Letting Go* (see Appendix B) for a specific case. The ritual is aimed mainly at one person: the partner requesting it after a breakup. It was discussed and evaluated in a KSA course (clinical pastoral training group) and first performed in 2018. Heinlein reports that he had looked for suitable templates but found nothing suitable. To meet this need, he himself developed a liturgical form for this situation. He shared his draft with the author of this article after a public Twitter request (Lauxmann 2022a). It is remarkable in several respects: Already in the opening prayer, the ritual release from the marriage vows is prepared liturgically with the active participation of one (ex-)partner. This is then ritually performed in a detailed liturgical sequence ("separation ceremony"). The ritual separation process involves first the promise given (reciprocally) at the marriage ceremony and then the actual marriage relationship. This declaratory act with its double thrust is performed "in all due respect for God", on the one hand, and "with God's help", on the other. The actual performative speech act is as follows: "You are released and loosed from your marriage vows"—thus, it is primarily related to the vow given. In the intercessory prayer, the present social network of the partner becomes publicly visible. The continuing relationship with God is not doubted for both persons who were formerly connected in a love relationship. It is undoubtedly a concise, personal, and practical ritual focusing on individuals whose (long-term) relationship has ended.

### 5.3. The "Three Pillar Model" (Austria 2019)

As part of the above-mentioned course in liturgics, a working group of four MA students presented a draft for an untitled end-of-relationship ritual built on three pillars (see Appendix C). Conceptually, the students note that their draft is a model not for a ritual of divorce but of *separation*. In doing so, they emphasize that a previous marriage is not a condition for participation: "Nowadays, not every couple [ . . . ] steps in front of the altar. Long and stable relationships can no longer be tied to a wedding ceremony, since they are also realized without marriage". Therefore, structural analogies to the wedding ceremony are deliberately avoided. The concept of guilt—which still plays a role in the Austrian

legal context of divorce today—is also deliberately avoided for this reason. Instead, they established three basic pillars for the ritual's structure in time and content: (a) lament; (b) commemoration/thanksgiving; (c) new paths and communion with God. The ritual is celebrated in a smaller prayer service (German *Andacht*), but in the public church service on Sunday, the pastor explicitly points out that such a celebration has taken place in the church, and the couple concerned is mentioned in the prayers of intercession (e.g., with the words "they have asked for the blessing" and "we are glad that they have taken this courageous step"). Thus, the ritual is not part of the regular Sunday service but is not completely separated from the life of the congregation either. It acquires a publicly visible dimension.

The course of this very elaborated ritual includes the following elements: (1) music; (2) greeting; (3) opening hymn; (4a) extended lamentation with symbolic ritual action (laying stones at the base of a tree); (4b) continuation of the symbolic action (harvesting the fruit of the tree); (5) hymn; (6) scripture reading; (7) short address; (8) mutual anointing of the couple; (9) blessing; (10) hymn; (11) prayer; (12) hymn; (13) agape celebration; (14) prayers of intercession; (15) Lord's Prayer; and (16) hymn. The aspect of lamentation is key in elements 1–4a. Good memories of the previous relationship together should come to light in the elements 4b and 5. The longest part of the ritual (elements 6–16) is dedicated to the new phase of life as separate individuals in continuing communion with God. The ritual even received media attention for its innovative character—reciprocal anointing is atypical for this ritual genre (Mörgeli 2020).

*5.4. The "Exploring New Paths" Retreat (Austria 2019)*

Five other students from this course on liturgics presented a practical proposal for Protestant retreats for separated and divorced people, entitled "*Exploring New Paths*" (see Appendix D). This proposal was worked out in such detail as to be ready for the initial implementation and testing. The design was based on a modular system in which various elements ("building blocks") can be replaced, expanded, and adapted. The students endeavored to ensure the greatest possible flexibility in implementation and a high degree of practicality. They even produced flyers and exercise instructions in an eye-catching design that could be made available to interested practitioners. The retreats are understood as an offer of the Protestant Church in Austria for groups of separated and divorced individuals. However, it is made explicit that participation is independent of religious confession or church membership. For the concrete implementation of the retreat days, it is recommended that participants form at least two groups, allowing former couples to take advantage of the offer independently of each other. Because the retreats are designed as a spiritual offering, they include Bible study, time together for silence and reflection, and time for group experiences in nature. Central are thematic workshop units on (a) *Lament*, (b) *Guilt, Forgiveness and Letting Go*, and (c) *Exploring new Paths*. This thematic combination of workshops is intended to show—based on a life-cycle ritual—a development perspective from grief/lament to hope. The retreat concludes with a service on the theme of "New Beginnings". In addition, another service is to be celebrated the following week in the participants' home region. At its presentation, the students demonstratively pasted their liturgical draft for the retreat group service and for the planned regional service into the official book of Protestant worship (*Evangelisches Gottesdienstbuch*) used by Austrian pastors. Thus, at the University of Vienna, a design for a divorce service already forms part of an official church book.

## 6. Discussion

The drafts developed at the University of Vienna show that the case study, a method refined at Harvard Business School, is also suitable as a tool for reflection in the humanities and practice-oriented theological production: it has generated innovative solutions to problems in the ritual and liturgical field of religious practice in a theological seminar context and strengthened the ritual competence of future theologians. In other words: the case study method can be productive of ritual.

Theoretical considerations in the field of ritual transformation and relevant distinctions (ritual adaptation, ritual transformation, ritual innovation, and ritual design; cf. Ahn 2012) are an important basis for informed theological ritual design in a field as sensitive as divorce rituals. Conversely, church practice aids on ritual design (e.g., Maier 2017) are relevant to research-based learning processes in academic educational contexts. What is intended for ecclesiastical and pastoral practice can also produce important insights for ritual theory.

The heterogeneity of these drafts testifies to the enormous innovative power of the younger theological generation. Moreover, a stronger decoupling from the standard church repertoire can be observed, although the link to genuine church contexts and parochial and clerical structures remains intact rather than being abandoned. By the same token, the drafts testify less to a tendency towards "dechurching" than to an expansion of the field of occasional church services, as recommended by practical theologians (Wagner-Rau [2000] 2008).

In terms of content, the drafts clearly indicate that the question of to whom a future offer of divorce rituals should actually be directed at is by no means trivial: possible addresses include individuals (one single ex-partner) and former couples (two former partners), as well as groups of individuals. What is also informative in this respect are two possibilities briefly discussed in the students' reflections on the three-pillar model: on the one hand, the witnesses to the former marriage (best man, maid of honor) are special persons in the close social environment of the main addressees and may, therefore, have a special ritual role; in addition, children (especially those of the dissolved relationship) must be adequately included and ritually addressed.

## 7. Conclusions

Divorce rituals offer an important opportunity for anyone looking to expand their skills in the field of ritual design, experiment with ritual processes, and learn more about the most undecided ritual fields within church culture. Divorce rituals may well meet with increased sympathy in theology and the church if their potential for enabling learning processes in the field of ritual design were recognized. It would be appropriate to no longer consider divorce rituals as marginal phenomena but to consciously bring them into focus in church and theological debates. It is high time that the mantra that there are no official divorce rituals be deleted from church websites. It seems at odds with the spirit of the gospels by restricting areas of action and experience. Moreover, and especially in view of what younger generations are proposing and how they position themselves theologically in ritual matters, the claim seems completely out of touch with our times. The article has shown that pioneers who speak out for divorce rituals in public, guerrilla practitioners who accompany affected persons in secret, and future theologians experimenting with ritual design in university settings have long challenged the oft-repeated Protestant mantra that "there is no official divorce ritual in the church". May their efforts not have been in vain.

**Funding:** This research received no external funding.

**Institutional Review Board Statement:** Not applicable.

**Informed Consent Statement:** Informed consent was obtained from all subjects involved in the study.

**Data Availability Statement:** Data is contained within the article and appendices.

**Acknowledgments:** Open Access Funding by the University of Vienna. Junior Researcher Funding by the Faculty of Protestant Theology, University of Vienna.

**Conflicts of Interest:** The author declares no conflict of interest.

## Appendix A

After familiarizing the students with statistical information on separation and divorce in Austria and providing them with liturgical literature and church agendas currently in use, the students were confronted with the particulars of the case. The case description presented in Table A1 was applied in the seminar in exactly this way (translated here by the

author). The case study followed the relevant phases of (a) confrontation, (b) information, (c) exploration, (d) decision, (e) disputation, and (f) collation (Brettschneider and Kaiser 2008; Friedrichsmeier 2017; Müller 2009).

**Table A1.** Case description structure.

| Case Structure Element | Full Description |
|---|---|
| 1. Decision to act in the Liturgical Committee | The Liturgical Committee of the Protestant Church in Austria decided in its last meeting that it would actively address the issue of "separation rituals and divorce services". This decision was made in consultation with the members of the Theological Committee who are in favor of the effort. There is no explicit mandate from the Synod to the Liturgical Committee. Such a mandate was last given for the "new wedding agenda" (agenda for the service on the occasion of a marriage (cf. EKiÖ-AB 2021)) and may well be considered common. As a rule, the Liturgical Committee acts on synodal mandate. However, it is also entitled to develop positions, contributions, or submissions independently of the synod and to present them for discussion, although so far this possibility has rarely been used. The members of the committee consider the topic of "separation rituals and divorce services" as well as the question of suitable liturgical offerings for separated or divorced persons, their children and their relatives to be important and therefore want to become liturgically active in this field. |
| 2. Decision on the course of action of the Liturgical Committee members | In addition to the decision to work on the topic, a proposal on the formal procedure was approved by consensus at the meeting: It was decided to develop and elaborate concrete liturgical considerations in working groups. The results of these working groups are to be presented in a next step. If necessary, they will also be discussed and voted on in the committee. The Liturgical Committee could, on the basis of a working group proposal approved by a majority, quickly prepare an official document by making a few clarifications and modifications, which would then be put on the agenda of the Synod. |
| 3. Points of discussion within the Liturgical Committee | At the committee meeting, however, not only were decisions made, but there was also open and sometimes fierce debate. One committee member noted that there is currently no real need among Protestants for separation rituals or divorce services; the experience is that there are very few requests. "Is this perhaps a completely unnecessary service that we are committing ourselves to, ultimately investing our time for nothing?" someone asked. The question of whether demand might increase if there was a visible, well-founded "product" in a church setting in future was also debated. A committee member pointed out that there were already colleagues who successfully performed rituals for divorced/separated people "in a pastoral setting". Finally, it was pointed out that there was a "somewhat unsatisfying" mention of divorce rituals in the FAQ on the website of the Protestant Church: "Is there an ecclesiastical ritual on the occasion of a divorce? One task of the pastor is to accompany people, including in crisis situations. Please discuss with your pastor which concrete forms and rituals make sense and are possible. So far, there is no official divorce ritual in the church". Because there is currently neither an agenda for a divorce service, nor an official template for a divorce ritual in a pastoral setting, nor a relevant church statement or contribution to the liturgical orientation of divorce rituals in the Protestant Church in Austria, and although the issue has not been silenced, the resolution to address the topic of "separation rituals and divorce services" was passed with an overwhelming majority. |
| 4. Wording of the resolution | The official resolution text of the Committee is as follows: "In order to support pastors in their task of assisting people in or at the end of separation and divorce scenarios, but also to close a gap in the field of liturgical practice, the Liturgical Committee will be addressing the topic of 'Separation Rituals and Divorce Services'. As of now, it will work intensively on concrete liturgical offers for separated and divorced persons and their relatives. By the end of November, we expect a number of working groups to have developed and presented concrete liturgical drafts for church 'divorce rituals' (or similar services) for further consideration. The aim of these drafts is to provide concrete support for the ritual or liturgical design of relevant practical cases, to clarify the proper setting (pastoral care/worship service/etc.), and to define the intended target group of such practices (separated/divorced couples, individuals, children, etc.) more precisely. A high level of liturgical concretion is intended. It is also envisioned that these drafts or practical proposals will be thoroughly discussed and, if necessary, voted on in the next meeting after they have been presented, so that—if necessary—a resolution can be passed based on these presentations. An official synodal draft may developed on the basis of work group results endorsed by the majority of members. An official document from the committee may serve to stimulate a discussion within the wider church and to objectify the debate about church divorce rituals from a liturgical perspective". |
| 5. Concrete task and time schedule | – You are a member of one of the working groups which is to prepare a liturgical draft or a practice proposal for the next meeting of the Liturgical Committee of the Protestant Church in Austria.<br>– On October 11 and 12, the first meeting of the members of your working group will take place. Other dates in October and November will follow. The proposal must be drafted by November 22.<br>– Marianne Fliegenschnee, chairwoman of the Liturgical Committee of the Protestant Church in Austria, * is not alone in looking forward to the presentation of the concrete elaboration of your draft on November 30. |

* In fact, Marianne Fliegenschnee was chair of the Liturgical Committee at the time of the seminar. She was also involved in the preparation of the seminar and officially invited to the presentation session of the students' proposals. Unfortunately, she canceled her attendance at very short notice.

## Appendix B

The following draft is entitled *Liturgy of Letting Go* and has been tested in church practice. The author is a pastor and was in his mid-forties when the draft was written. He comments: "The design is of my own devising, there was nothing suitable to be found".

His draft was discussed collegially and professionally reflected upon in a clinical pastoral care training class (KSA-Kurs).

**Table A2.** Structure of the *Liturgy of Letting Go*.

| Liturgical Element | Spoken Text |
|---|---|
| 1. Trinitarian formula | In the name of the Father, and of the Son, and of the Holy Spirit. Amen. |
| 2. Welcome | We are here because __________wants to divorce her marriage before God today and to be released from her marriage vows that she made in this church on __________. |
| 3. Psalm | *(Psalm 121, read together from the hymn book)* |
| 4. (Opening) prayer | Let us continue to pray:<br>Lord our God,<br>we come to you today<br>to resolve a promise.<br>You know the strengths and weaknesses of us humans.<br>You know it, because you yourself have lived among us.<br>We ask you today for forgiveness<br>for everything that has not worked out in our lives.<br>Above all, we bring before you the marriage vows of __________,<br>which she could not keep.<br>In sincere repentance we ask:<br>Forgive us and give us a new beginning.<br>Give us anew the power of your Holy Spirit.<br>Renew our hearts and minds.<br>Amen.<br>If this is also your request, answer "Yes."<br>[ __________: *Yes*.]<br>Merciful God<br>forgive you all your sins in Jesus Christ.<br>What has been shall no longer burden you.<br>What is to come shall not terrify you.<br>God's grace set you free<br>for a new beginning with him and with your fellows.<br>Amen. |
| 5. Start of the separation ceremony | Let us go before the altar.<br>[*Walking to the altar together.*] |
| 6. Detachment question | I ask you, __________<br>Do you want to detach yourself from your promise of __________ and separate from __________ as your husband in all due respect for God, then answer:<br>Yes, with God's help.<br>[ __________: *Yes, with God's help.*] |
| 7. Detachment | Jesus Christ says:<br>I will give you the keys of the kingdom of heaven,<br>and whatever you bind on earth<br>will be bound in heaven,<br>and whatever you loose on earth<br>will be loosed in heaven. (*Mt. 16:19*)<br>I claim this today and speak:<br>You are released and loosed from your marriage vows.<br>Amen. |

**Table A2.** *Cont.*

| Liturgical Element | Spoken Text |
|---|---|
| 8. Blessing (I) | Almighty God bless you<br>and give your heart all freedom.<br>May the peace of Christ dwell in your heart<br>and in your house.<br>May whoever is in need find comfort and help in you.<br>May good friends stand by you in joy and sorrow,<br>and the blessing promised to the merciful<br>come upon thy house.<br>God speaks to you today:<br>For the mountains may depart<br>and the hills be removed,<br>but my steadfast love shall not depart from you,<br>and my covenant of peace shall not be removed,<br>says the LORD, who has compassion on you. (*Isa. 54:10*)<br>Amen. |
| 9. Prayer of intercession | Lord,<br>we have heard how ____________ has detached herself from her vows<br>and how she was released from her former marriage.<br>We humbly ask:<br>She gives her former husband ____________ back to you today<br>and says goodbye in her heart.<br>Keep him in your love and give him your blessing.<br>We ask you today for all the people<br>with whom is associated today:<br>Give her a new and open heart<br>for her partner and for his children,<br>for her father and her mother ____________,<br>for her friends.<br>Bless these relationships in which she stands,<br>and the people with whom she unites in love. |
| 10. Silence | We now bring our own prayers before God in silence.<br>[*Silent Prayer*] |
| 11. Lord's Prayer | Let us now pray as Jesus taught us and as Christians do throughout the world: [*Bells ringing*]<br>[*Lord's Prayer, spoken together*] |
| 12. Blessing II | Go into this day of freedom with the blessing of God:The LORD bless you<br>and keep you;<br>the LORD make his face to shine upon you,<br>and be gracious to you;<br>the LORD lift up his countenance upon you,<br>and give you peace.<br>Amen. |

## Appendix C

The following ritual proposal was created by a group of students at the University of Vienna in the winter semester 2019/20. The students discussed the draft together and with experts. They also wrote a liturgical essay on it, in which they comment on the draft and also formulate individual modification proposals to the group document.

**Table A3.** Structure of *The Three Pillar Model*.

| Phase | Liturgical Element | Spoken Text |
|---|---|---|
| A | Music | – |
| A | Welcome | [*No suggested text given*] |
| A | Opening hymn | – |
| A | Psalm of lamentation | [*Reading from Ps 27, Ps 143 (excerpts) optional*] |
| A | Symbolic action I (liturgical lament) | [*A tree with fruit stands in the altar area; stones are prepared for lamentation.*] Your sorrow and sadness are so great that you do not know where to put them. God, I painfully miss what is no more. But I no longer rebel against it. We lay down our sorrows and our pains as stones that have burdened us in the past –to you, God. [*Request to lay down the stones*] [*Music and laying down the stones by the participants*] |
| B | Symbolic action II | We now go back to the beginning of your common history, to the good fruits of our life together. Gratefully we look back on the good things. Despite the sorrow, these memories are not lost for us. [*Music and harvesting of the fruits by the participants*] |
| B | Hymn | [Two specific hymns from a hymnal of the Bavarian Institute of Worship are suggested: "*Wie ein Fest nach langer Trauer*" (025) and "*Du bist mein Zufluchtsort*" (032)] |
| C | Scripture reading | [Suggested are "*The Separation of the Elements*" in the Creation Story (Gen 1) or "*The Parting of Abram and Lot*" (Gen 13).] |
| C | Short sermon | [*The focus of the sermon is change as a step toward the new order of life. It is based on Gen 1 or Gen 13*] [*End of sermon:*] God has given us the gift of creating something new out of what has been separated, he encourages us to let it go as it was before in order to create something new. |

**Table A3.** *Cont.*

| Phase | Liturgical Element | Spoken Text |
|---|---|---|
| C | Mutual anointing | [*Text read out aloud by the pastor:*]<br>"To release this person,<br>to whom my heart is attached,<br>who is completely irreplaceable to me,<br>for his own path –<br>O Lord, that is so unspeakably difficult.<br>A harder loss could not have hit me.<br>To let go of this person<br>with whom I could share the innermost,<br>who made me rich like no other.<br>I don't know how I can manage<br>to accept the hard reality.<br>Everything is incomprehensible to me.<br>My God, stay with me.<br>Do not leave me alone with my dismay".<br>[*Text spoken by the celebrant:*]<br>In the midst of all sadness and gratitude, I would like to promise you that God will be by your side on your onward journey.<br>[*The separated draw the sign of the cross on each other's forehead with oil and say the following words of interpretation to each other:*]<br>I anoint you with this oil as a sign of God's love. May it be a balm for your soul and ease your pain. Let it grow new in you, so that you can go your way with confidence. |
| | | [*Additional liturgical options for related children:*<br>(A) *Cave building*: Children can build a cave in the church before the service, if the service takes place there, and can also withdraw there during the service (materials: chairs, blankets, and cushions).<br>After the anointing, the parents may enter the cave with the celebrant and bless the children together. Afterwards, everyone leaves the cave together and returns to the altar area. (B) *Carrying a cloth*: The cloth is placed on the altar at the beginning of the prayer; after the anointing, the children are called to the front; one child lies down on the cloth and the parents (possibly with relatives) lift the child up in the cloth for a few minutes.] |
| C | Blessing | God bless you with the love that makes you alive, that you may let go of disappointment, learn to start anew, and go your way upright. Amen |
| C | Hymn | [A specific hymn from a hymnal of the Bavarian Institute of Worship is suggested: "*Geh unter der Gnade*" (0116)] |
| C | Prayer | God, sometimes I still feel trapped in the past and not yet alive again. But You strengthen us on our way. You give us food and drink. You show us that we can sit together at Your table. |
| C | Hymn | [A specific hymn from a hymnal of the Bavarian Institute of Worship is suggested: "*Wenn das Brot, das wir teilen*" (091)] |

**Table A3.** *Cont.*

| Phase | Liturgical Element | Spoken Text |
|---|---|---|
| C | Agape ceremony | [*A table with bread and wine/grape juice is already prepared in the room at the beginning of the prayer, but it is still covered with a cloth. The celebrant invites all those present (including friends and relatives) to the table.*]<br>Come, God invites us to eat and drink together at his table.<br>[*The cloth is taken from the offerings.*]<br>As this bread, made from many grains, is one bread, and as this wine, made from many grapes, is now one drink, so God wants to bring us humans together—in this community and on the whole earth. Come and eat of this bread that unites us. Together we sit at God's table as separated ones, as his children. Give one another of the bread and wine, the Lord wants to strengthen us for our journey.<br>[*Bread is broken and shared with each other (possibly musical accompaniment).*]<br>Together we have shared bread and wine, as Jesus once did with his disciples. Now join hands as a sign of peace.<br>[*All join hands.*]<br>Lord, we thank you, you have put people who have brought their sorrow, their worries, and their gratitude before you, together at one table to give them food for the journey. Strengthened by you, we go on.<br>Amen [*spoken by all*]. |
| C | Prayers of intercession | [*The prayers of intercession are formulated and spoken by the relatives. During these and the Lord's Prayer, everyone remains at the table but stands to pray.*] |
| C | Lord's Prayer | [*The Lord's Prayer is sung or spoken. For the sung variant, a specific song from a songbook of the Bavarian Institute of Worship is suggested: Unser Vater (KAA).*] |
| C | Aaronic Blessing | [*Spoken by the pastor in the traditional form.*] |
| C | Closing hymn | [*The recommended hymns are from the Protestant hymnal widely used in Germany and Austria: "Bewahre uns" (EG 171) or "Wo ein Mensch Vertrauen gibt" (EG 648). After the song, those present leave the church, possibly with musical accompaniment. Optional shared meal to follow.*] |

**Appendix D**

The following proposal with ritual aspects was created by a group of students at the University of Vienna in the winter semester 2019/20. The students discussed the draft together and with experts. They have also written a liturgical essay on it, in which they comment on the draft and also formulate individual proposals for modifying the group document.

**Table A4.** Structure of the *Exploring New Paths* model.

| Friday Program | | |
|---|---|---|
| 5 P.M.–6 P.M. | Arrival | |
| 6 P.M.–7 P.M. | Dinner | |
| 7 P.M.–8 P.M. | Get to know | This unit is designed to focus on the situation of the participants. They come from different settings, separation situations, family constellations, and daily routines. The evening is designed to make the participants aware that the coming weekend is just for them and represents a contrast to their everyday lives. It is agreed that in a protected setting all personal information will remain secret. A spiritual impetus, supported by a song/hymn, a picture or a movie clip inaugurates the shared retreat. |

**Table A4.** *Cont.*

| | | |
|---|---|---|
| **Friday Program** | | |
| Saturday program | | |
| 8 A.M. | Breakfast | |
| 9 AM–1 P.M. | Workshop 1: Lament | Bibliologue (Germ. *Bibliolog*) * <br> The workshop on lamentation starts with a reading and discussion of Job 19:7–19 or Lamentations 5. It should become clear that lamenting is a genuinely human behavior, which serves the articulation of pain, grief, and suffering and need not be connected with feelings of shame. It will be shown that lamenting is also an expression of piety deeply rooted in the Bible, since those who lament turn to God with their problems. |
| | | Lament (individual) <br> The scripture reading is followed by an opportunity for participants to give voice to their own lamentations. Participants can either pray, reflect on their lament, or write their lament down in a devotional room using pen and paper. |
| | | Lamentation wall <br> Written thoughts will be placed symbolically on a prepared lamentation wall. |
| 1 P.M.–2 P.M. | Lunch | |
| 2 P.M.–6 P.M. | Workshop 2: Guilt, Forgiveness, Letting Go | Walk with God <br> The participants set out alone with a booklet and pen on a 1.5 hour *walk with God*. The walking route is printed in the booklet. <br> Stations for thought <br> The booklet suggests four stations for thought: <br> Station 1: Think about heavy feelings. Find a symbol (e.g., a suitable stone) for these feelings. Station 2: Reflect on the experience of not being able to change or control things. You are invited to bring these feelings before God in prayer. (Psalm 13 as inspiration.) Station 3: Allow feelings of failure and self-reproach. Invitation to bring self-reproaches before God in prayer. (Quote from John Newton as inspiration: "I have learned two important things in my life: that I am a great sinner and that Christ is an even greater Savior".)/Station 4: Reflecting on the walk in writing, wrapping the notes around the stone/symbolic object. [*The wrapped items are (later) released in a joint prayer service.*] |
| | | Joint prayer <br> When all participants have gathered, one by one, the symbolic objects are put down in the center of the circle, where a candle is lit. The participants have the opportunity to formulate a request or thanks to God aloud before saying the Lord's Prayer. Afterwards, there is time for reflection. |
| 6 P.M.–7 P.M. | Dinner | |
| 7 P.M.–8 P.M. | Workshop 3: Exploring new paths | Scriptural inspiration <br> The workshop resumes with a chance to reflect on Jeremiah 1:4 to 7:10–14. |
| | | Postcard writing <br> The participants write themselves a postcard, which they will receive one year later. Their own future perspective can be formulated here in concrete terms. |
| | | Sharing experiences and trust games <br> The group shares experiences. Trust games are played to internalize that no one is left alone with their separation experience. |
| | | Service preparation <br> The workshop unit will end with the preparation of the worship service. The participants formulate prayers of intercessions or make candles as symbols of light and hope in difficult life situations. Finally, the experiences are reflected in the group. |
| | Campfire | The campfire serves as a cozy gathering and an end to the day. The evening is intended to be informal. |

**Table A4.** *Cont.*

| | **Friday Program** | |
|---|---|---|
| | Sunday program | |
| 8 P.M.–9 A.M. | Breakfast | |
| 9 P.M.–10 P.M. | Reflection & Sun Shower | Reflection/feedback<br>Before the weekend ends with a worship service, there will be time for reflection on the retreat. Feedback should also be sought on the accommodation, catering, workshops, guidance, and environment. |
| | | Sun shower<br>In the "Sun Shower", all participants draw a sun in the center of a sheet of paper and write their name in it. The rays of the sun are drawn to the edge of the sheet. Now the sheet is passed around and each participant writes a message on it. At the end, everyone is allowed to take their own sun home. |
| 10 P.M.–11 A.M. | Closing service | Formally, the closing service (on the 8th Sunday after Trinity, if possible) will follow the traditional liturgy, but there must be *an extended element of intercession*, in which intercessory prayers written by the participants can be brought before God together. *Individual blessings* should also be offered if agreed and announced before the service, and be given close to the altar. Participants who request an individual blessing are invited to come forward one after the other and receive a blessing from the pastor. The *Lord's Supper* is to be celebrated with all participants. |
| | Departure | |
| | Additional program | |
| regular time | Regionalworship service | On the *following Sunday* (the 9th Sunday after Trinity), the regular Sunday service will be held under the motto "Breaking New Ground" to connect the retreat groups with the congregation (and vice versa). It should be designed as a service especially for separated and divorced people, their families, relatives, and friends. This themed service is to be advertised throughout the region. |

\* Bibliologue is an *independent homiletic concept* that is close to the reader-reception approach. It is represented in German-speaking countries by *Uta Pohl-Patalong*. English-speaking readers will find a brief summary in the presentation manuscript *Bibliologue. An Introduction in Theory and Practise* (Pohl-Patalong 2012). There are several monographs on the subject (esp. Pohl-Patalong and Aigner [2009] 2013a, 2013b; Pohl-Patalong 2019).

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
