# Peer review of "“There Is No Official Divorce Ritual in the Church”—Challenging a Mantra by Ritual Design"

_religions, doi:10.3390/rel14020137_

Round 1
Reviewer 1 Report
This is a nicely structured and well-written article on the state and possibilities of divorce rituals in the Protestant tradition. It puts the finger on a much-needed practice in the church and presents a few options that can be used as future crisis rituals. I especially like the example of separation after a longstanding relationship (not marriage).
The article is written in a high standard of English, and no typos were identified.
Author Response
"Please see the attachment."

Reviewer 2 Report
This is a clearly written, well structured account of three proposals for a liturgy or ritual to mark a divorce in the context of the general absence of such services after divorce in German-speaking Protestant - evangelical - Lutheran churches in Germany, Austria and Switzerland. As someone with experience in such services/rituals in an English-speaking context, especially Anglicans, I found it most interesting and illuminating - and I did not know anything about the situation in German Protestant churches, so I am very grateful and support its publication. However, I have three main points in response:
First, the article is predominantly descriptive, rather than analytical or
critical. The only critical discussion comes in the attempt to ascertain the origins of the "mantra" often repeated on church websites that “There is no official divorce ritual in the (protestant) church” with regard to its authorship and possible date (2012-Barbara Driessen? 2007/2008-Margot Käßmann?). There is also some analysis of the contrast between church involvement in social rituals after a major disaster (widely welcomed in public), compared with the rituals for birth, marriage and death (just regular routine) and rituals following divorce (almost completely missing). There is some attempt to discuss possible reasons for this contrast - but much more could be said, and the variation is interesting and thought-provoking.
Second, the concentration of the article, its discussion and bibliography is
entirely on German-speaking Protestantism. Since I was previously ignorant of this, I found it most interesting and thought-provoking - and it seems to me that its claim and merit for publication rests entirely on bringing this area into wider scholarly notice - hence the predominance of high/positive boxes ticked above. The "average"/"can be improved" boxes ticked represent my feeling that a wider discussion involving such services and rituals with English-speaking Protestant/Anglican traditions (or even better, also including Catholic thinking about dispensation from vows, for instance) would turn this already interesting and good piece into an excellent and original challenge - but that is clearly outside the bounds of the author's intention, and possibly the journal's remit? I personally would very much like to compare and contrast the German material in this article with some English services - for example, those which can be found in Pastoral Prayers: Liturgies and Blessings for Health and Healing by Tess Ward (Canterbury Press, Norwich, 2012) pp. 98-102 and 204-208; Pastoral Prayers: A Resource for Pastoral Occasions, ed. Richard Deadman, Jeremy Fletcher, Janet Henderson, Stephen Oliver (Mowbray, 1996), pp. 47-53; Daring to Speak Love’s Name, Elizabeth Stuart (Hamish Hamilton, 1992), pp. 95-100; see also Vows and Partings (The Methodist Church, 2001).
Third,the author’s descriptive approach merely tells us what happens in the three examples which he describes. There is no theological or psychological analysis of the various rituals, prayers, words, or actions undertaken – such as how can solemn lifelong vows be set aside, or broken promises handled, let alone how grieving hears or guilty feelings might be healed. Some discussion on the authority of the church or its leaders / pastors / priests to pronounce forgiveness in the name of God would have helped to assess these services – both theologically, and also from the point of pastoral or psychological effectiveness. Once again, for me, this would transform this already interesting and good piece into an excellent and original challenge - but that is clearly outside the bounds of the author's intention, and possibly the journal's remit?
I have no hesitation in recommending the publication of this article as a good example of what it is – a recognition of the absence of such liturgies/rituals/services after divorce in the German-speaking tradition and a descriptive account of several attempts by younger leaders to remedy this lack. This is sufficient to merit wider publication and interest on its own.
However, you can also see how this article has spoken to me and to my interests, and raised important questions for me – and how much more could be achieved than mere description of German experiments by increasing its remit and field to include English sources, together with more theological, pastoral, and psychological critical analysis.
Author Response
"Please see the attachment."

Reviewer 3 Report
This is an excellent article that addresses a significant contemporary ritual practice in a creative and concrete manner. In addition to an interesting content analysis of how divorce ritual is presented on church websites and valuable examples of recently developed ritual practices for marking the end of a significant relationships, the article makes a pedagogical contribution by modeling the use of a business school style extended case study for teaching about ritual.
The suggestions below are minor recommendations for the author to consider:
- It could be helpful to offer a couple of sentences about the social context of divorce in the geographic regions that are the focus on this research (legal aspects, societal norms, attitudes in Protestant Christianity, etc.). The case study implies this type of information was provided to students. It may also benefit readers.
- The introduction under “The unequal focus on rituals in theology and the church” (page 2) could more clearly outline the three categories that follow. I was able to understand this paragraph only after looking at the headings below.
- The metaphor of “unloved stepchildren” on page 3 could be reconsidered, especially to be sensitive to readers with children who have experienced divorce.
- In the quotations from websites, there is a dual emphasis: 1) the lack of official divorce rituals that is the focus of the article, and 2) the “important task of pastors to accompany people in crisis situations.” The first point is the focus of the article to the neglect of the second. A stronger emphasis on and deeper exploration of the second point about pastoral accompaniment and the nature of divorce as a crisis could strengthen the case for separation rituals.
- Presumably the information from the websites has been translated for use in this paper. Including the original language(s) for the key phrases under discussion could be beneficial.
- In the appendices, there are a couple of words not usually used in English “votum” (page 14) and “bibliologue” (page 19). Perhaps these have been improperly translated?
- Due to anonymization, it is unclear whether the names of the students who created these rituals are included or only the name of the class and institution. With their permission, I would strongly encourage including their names due to their significant contributions to this article.
Thank you for this practical theological contribution to liturgical and ritual scholarship.
Author Response
"Please see the attachment."

Reviewer 4 Report
This is a very useful essay that makes the case for the importance of divorce rituals in Protestant churches and provides very helpful examples.
The rituals (or liturgies) provided and the discussion of them will be very helpful to all who are interested in this topic, it is great that it is being published in English. The focus on Protestants in Germany, Austria, and Switzerland is entirely appropriate and explains why all the secondary sources that are explicitly about divorce rituals are in German. There are some English sources on this topic, however. Mentioning them would be helpful for readers who do not have German.
Just for your information: I believe one of the first denominationally published divorce rituals in the United States was in 1976 by the United Methodist Church in Ritual in a New Day: An Invitation, vol. 4, Supplemental Worship Resources (Nashville: Abingdon Press, 1976). It became a topic of scandal and the denomination, to my knowledge, has not provided any divorce liturgy (ritual) since then. (This publication was simply of "supplemental" non-official services, but it was by an official church body.)
Regarding one of the author's main points: it is interesting to see how similar the language about divorce rituals is across the different groups, particularly the off-repeated phrase "there is no official divorce ritual." But as an American with only limited knowledge of European German-speaking churches, I may not be fully understanding why it is so important to trace out. It seems quite clear that the various organized church bodies do not have an official divorce ritual, so the statement just seems factual. If the statement implies that there can be no official divorce ritual in any Protestant church, then that should be made clear.
I do accept that the author's main point is that even if there is no official ritual, that statement does not need to be repeated in statements that suggest that individual clergy may supply such rituals.
The English is excellent and easy to understand. But there are some odd (non idiomatic) word choices and a few sub-par sentence instructions.
The category "unloved" is clear, but unusual in this context. I can't think of a perfect synonym, however. "Not respected" is the best I can think of, but may not be right. The second sentence of the abstract should be broken in two. The first comma (after "day") should be made into a period and the rest of the sentence formed into a new sentence beginning "This is despite the fact that there are numerous impulses . . ."
Other language notes:
In the second paragraph of the introduction change "first step," "second step" etc. to "first part," "second part" etc.
on page 2, "hyped" is too colloquial a word, I think. "valued" is what I think would be better.
on page 3 "lethargically" sticks out as an odd word "defeatedly" seems better to me.
also on page 3 "This argument is played up and down" -- I'm not quite sure this means. Does it just mean "This argument is constantly emphasized"?
Author Response
"Please see the attachment."

Round 2
Reviewer 2 Report
My original review said that I found it most interesting and illuminating - and I did not know anything about the situation in German Protestant churches, so I am very grateful and support its publication. However, I had three main points in response:
"First, the article is predominantly descriptive, rather than analytical or critical.
Second, the concentration of the article, its discussion and bibliography is entirely on German-speaking Protestantism.
Third, the author’s descriptive approach merely tells us what happens in the three examples which he describes. There is no theological or psychological analysis"
The revised version has not changed anything to do with the above three points, nor have they included the English sources which I recommended. Therefore I have not changed any of the bullet point boxes ticked - they are as previously.
I concluded previously:
"I have no hesitation in recommending the publication of this article as a good example of what it is – a recognition of the absence of such liturgies/rituals/services after divorce in the German-speaking tradition and a descriptive account of several attempts by younger leaders to remedy this lack. This is sufficient to merit wider publication and interest on its own.
However, you can also see how this article has spoken to me and to my interests, and raised important questions for me – and how much more could be achieved than mere description of German experiments by increasing its remit and field to include English sources, together with more theological, pastoral, and psychological critical analysis."
Since nothing relevant to my review has changed, I remain of the same opinion - it can be published as it stands, but it is a missed opportunity for theological and critical analysis, as well as comparison with English-speaking sources.